# Layover Intermediate Layer for Multi-Label Classification in Efficient Transfer Learning

**Seongha Eom**   **Taehyeon Kim**   **Se-Young Yun**
KAIST AI
{doubleb, potter32, yunseyoung}@kaist.ac.kr

## Abstract

Transfer Learning (TL) is a promising technique to improve the performance of a target task by transferring the knowledge of models trained on relevant source datasets. With the advent of advanced depth models, various methods of exploiting pre-trained depth models at a large scale have come into the limelight. However, for multi-label classification tasks, TL approaches suffer from performance degradation in correctly predicting multiple objects in an image with significant size differences. Since such a *hard instance* contains imperceptible objects, most pre-trained models lose their ability during downsampling. For the *hard instance*, this paper proposes a simple but effective classifier for multiple predictions by using the hidden representations from the fixed backbone. To this end, we mix the pre-logit with the intermediate representation with a learnable scale. We show that our method is effective as fine-tuning with few additional parameters and is particularly advantageous for *hard instances*.

## 1   Introduction

Transfer learning  (TL) methods have been developed into two categories: *finetuning* and *linear probing*. While finetuning aims to fully optimize pre-trained network for target task, linear probing trains only a head classifier while freezing the backbone network [19, 20, 10], which they share a trade-off between efficiency and optimization. Nevertheless, both methods show great success in downstream tasks such as object detection [3], segmentation, and classification [24, 18]. Despite the promise of TL, multi-labeled classification (MLC) task remains an arduous challenge since hidden representations from existing pre-trained networks (e.g., ResNet or Swin-Transformer) must capture a wide range of objects, from tiny imperceptible objects to large clear objects.

To address this challenge, we first conduct a pilot experiment to take a look at images with low prediction scores, which are cluttered with small imperceptible objects and large distinct objects, termed as *hard instances*. Here, the pre-trained network is linear probed with MS-COCO dataset [22] (The concrete process is reported in AppendixA.1). Interestingly, these *hard instances* frequently fail to be predicted and have visual commonalities regardless of class in that they contained multiple objects of various sizes, including small cluttered objects. This observation leads us to three insights: (1) lower class-wise performance has no relationship with class size or model architecture, (2) lower class-wise performance can be attributed to containing more hard instances than others, and (3) the representation obtained from the pre-trained network is imperfect.

In this paper, we present a simple but effective method by using intermediate representations as hints in knowledge transfer inspired by the works [36, 30]. We propose a new representation summarization (RS) module, which prevents the loss of spatial information from imperceptible objects by properly transforming it into a usable form (Figure 1 (A)). Subsequently, we mix a RS module with the feature extracted from the backbone with an adequate portion. More precisely, we use an adaptive mix-up scale which learns from the pre-logits (Figure 1(B)). Hidden representations printed during propagation can be good sources for model training. We show that mixing the two representations

Has it Trained Yet? Workshop at the Conference on Neural Information Processing Systems (NeurIPS 2022).

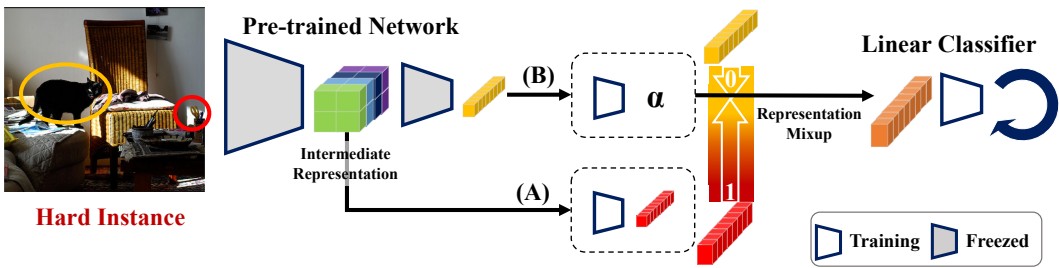

Figure 1: Pipeline overview: **(A) Representation Summarization (RS)** module refines intermediate representation. **(B) Scale Learning (SL)** layer learns a proper ratio using a feature extracted from a pre-trained network. Processed representation from (A) is properly mixed with the extracted feature by the proportion predicted by (B). Hence, the mixed output is passed to the linear head classifier for training.

gives comparable performance to fine-tuning where the scale value indicates the level of hardness of instance.

## 2 Method: Intermediate Representation as a Hint

In order to use intermediate representation, two questions need to be answered: how to use it with feature output extracted from the backbone and how much it involves training the classifier. The following subsections describe two auxiliary branches that address the above requirements.

### 2.1 Representation Summarization (RS)

For a network of $L$ layers, an intermediate representation at the $l$-th layer, $z_l \in \mathbb{R}^{H \times W \times C}$, is expressed as follows:

$$z_l = \boldsymbol{W}_l \cdot h_{l-1} + \boldsymbol{b}_l \quad ; \quad h_l = \phi(z_l) \tag{1}$$

where $\boldsymbol{h}_0 = \boldsymbol{x}$ is the input, $\phi$ is the activation function, and $\boldsymbol{z}_L \in \mathbb{R}^N$ is the final output from given network.

Vision models learn to summarize input into a dense representation with downsampling operation. More specifically, linearly embedded token [24] or feature map [18, 14] shrinks with downsampling operation as channel size $C$ increases. In order for the intermediate representation to be dense and informative enough, we adapt the perspective from Hu et al. [15], which views a feature map as a collective of local descriptors whose summarized statistics express input. Thus, the Representation Summarization (RS) module is composed of 2d pooling to reduce channel-wise statistics and MLP to learn inter-dependencies between channels. The representation transformed by the MLP layer is activated by sigmoid throughout all experiments, but any other activation function can be applied. The transformed representation is denoted by

$$\hat{z}_l = \text{MLP}(\text{Pool2d}(\boldsymbol{z}_l)). \tag{2}$$

**Note.** For patch representation in works such as [24], the spatial dimension is already mixed in the patch merging step. Therefore, we directly flatten the representation and apply pooling 1d.

$$\hat{z}_l = \text{Pool1d}(z_l')) \tag{3}$$

where $z_l'$ denotes flattened $z_l$.

### 2.2 Representation Mixup (RM)

Previous works either use representation extracted from Yim et al. [36] or use weights from layers of pre-trained models by simple averaging [34] or partial selection algorithm [10]. Along the line of this belief, intermediate representation can act as hints for capturing hard false negative samples with simple aggregation methods such as weighted sum or concatenation. The most straightforward method is to mix them in proper portion. We are inspired by several works on data augmentation using interpolation for mixing up in input space or feature space for better representation learning [39, 31]. An interpolation of hidden representations $z_i, z_k$ with respect to $z_i$ is expressed as

$$z' = z_i + \alpha(z_k - z_i) \tag{4}$$

Table 1: Mean average precision (mAP) and various metrics of transfer learning methods (FT: full fine-tune, RM: representation mixup (proposed), LP: linear probe) and previous state-of-the-art architectures reported from [6, 2, 5] on MS-COCO. The number of parameters are marked $\ll 1$, $<1$ if less than 0.5M and 1M respectively.

| Methods | | Resolution | Params (M) | mAP | CP | CR | CF1 | OP | OR | OF1 |
|---|---|---|---|---|---|---|---|---|---|---|
| ML-GCN [6] | | 448 x 448 | 46 | 83.0 | 85.1 | 72.0 | 78.0 | 85.8 | 75.4 | 80.3 |
| Tresnet-l [2] | | 448 x 448 | 55 | 86.6 | 87.2 | 76.4 | 81.4 | 88.2 | 79.2 | 81.8 |
| MlTr-m [5] | | 384 x 384 | 62 | 86.8 | 84.0 | 80.1 | 81.7 | 84.6 | 82.5 | 83.5 |
| Pre-trained Network | TL method | | | | | | | | | |
| | FT | 384 x 384 | 195 | 74.9 | 83.4 | 46.4 | 64.9 | **88.0** | 54.6 | 70.6 |
| Swin-L | **RM** | 384 x 384 | $\ll 1$ | **81.0** | **85.1** | **68.7** | **76.1** | 81.3 | **72.4** | **76.6** |
| | LP | 384 x 384 | $\ll 1$ | 76.9 | 84.3 | 60.7 | 70.5 | 87.7 | 64.0 | 74.0 |
| | FT | 384 x 384 | 87 | 73.4 | **84.4** | 43.3 | 60.7 | **89.1** | 51.7 | 68.0 |
| Swin-B | **RM** | 384 x 384 | $\ll 1$ | **79.9** | 83.9 | **67.5** | **75.0** | 85.8 | **68.8** | **76.8** |
| | LP | 384 x 384 | $\ll 1$ | 72.9 | 83.9 | 47.6 | 60.7 | 88.6 | 55.4 | 68.2 |
| | FT | 448 x 448 | 387 | **79.4** | 85.5 | 64.9 | 73.9 | 85.3 | **68.6** | 76.1 |
| ResNet101x3 | **RM** | 448 x 448 | 5 | **79.4** | **87.0** | **65.1** | **74.4** | 86.9 | 67.9 | **76.2** |
| | LP | 448 x 448 | $<1$ | 77.5 | 84.9 | 62.0 | 71.7 | **87.0** | 65.3 | 74.6 |
| | FT | 448 x 448 | 216 | 75.1 | 76.4 | 36.7 | 49.6 | 82.1 | 46.5 | 59.3 |
| ResNet50x3 | **RM** | 448 x 448 | 5 | **79.3** | **85.4** | **64.6** | **73.5** | **86.5** | **67.6** | **75.9** |
| | LP | 448 x 448 | $<1$ | 75.6 | 83.5 | 61.1 | 71.2 | 84.3 | 64.2 | 73.4 |

where $\alpha$ denotes the scaling factor.

Given pre-trained backbone $f$, newly initialized linear classifier $g$, and input image $x$, the final output score $s$ is $s = g \circ f(x)$. The final output representation from the pre-trained backbone is $f(x) = z_L$. The pre-trained backbone can be expressed as two parts $f(x) = f_2 \circ f_1(x)$ and intermediate representation $h$ can be retrieved while passing partial backbone $f_1$. The final representation $z'_L$ passed to classifier $g$ can be retrieved as follows:

$$h = f_1(x), \qquad \hat{h} = \text{RS}(h) \in \mathbb{R}^N, \qquad z'_L = \hat{h} \cdot \alpha + f_2(h) \cdot (1 - \alpha) \tag{5}$$

In this paper, following the recent works, we design a learnable parameter $\alpha$ by summarizing the hidden representation into a single value between 0 and 1 as

$$\alpha = \sigma(\boldsymbol{W} \cdot z_L + \boldsymbol{b}). \tag{6}$$

While previous mixup-based works [41, 13] use a beta distribution for $\alpha$, here we use an auxiliary linear layer to learn the level of mix in an instance-wise manner.

## 3 Experiments

**Result** Experiments are conducted on MS-COCO [22] and Pascal-VOC [11]. The proposed method RM is compared with conventional TL methods, i.e., to linear probe (LP) and to full fine-tune entire network (FT). As shown in Table 1, the number of parameters involved in training of RM is drastically smaller than that of any other comparisons [5, 2, 6] as well as FT. With the small number of parameters that involve in optimization, the proposed method gives comparable or better performance to fine-tuning. Also, note that our method requires much less training. Models in comparisons [5, 2, 6] require 100, 80, and 200 epochs respectively to achieve the desired performance, while ours show comparable results in 30 epochs. Details for setting are in AppendixA.2. Metrics show that the proposed method has the most gain in recall (CR, OR), meaning that the method can identify false negative hard instances as true positives. All the results from RM in Table 1 are of representation extracted from the first block (RM@1). Comparisons between different blocks are reported in Table2a, 2b.

**Activation Maps** Figure 2 shows activation maps of samples in MS-COCO with ResNet101x3 trained with proposed method with representation extracted from 1st block. Here, we use an EigenCAM method [25] for printing the activation maps. The samples are all true positives and show top 3 highest scores within each class when inferred by the aforementioned model while had showed worst 3 lowest scores when inferred by baseline (i.e., linear probing). The labels which class

Table 2: MAP of intermediate representation mixup extracted from different blocks (@1, @2, @3) compared with other methods, LP, and FT.

(a) MS-COCO dataset.

|  | ResNet101x3 | ResNet50x3 | Swin-B | Swin-L |
|---|---|---|---|---|
| LP | 77.53 | 75.38 | 72.30 | 76.68 |
| FT | 79.40 | 79.40 | 73.44 | 74.88 |
| RM@1 | 79.44 | 79.31 | 79.89 | 81.01 |
| RM@2 | 79.47 | 79.14 | 79.90 | 80.76 |
| RM@3 | 79.45 | 79.22 | 78.91 | 80.82 |

(b) Pascal-VOC dataset.

|  | ResNet50x3 | Swin-B |
|---|---|---|
| LP | 93.7 | 91.6 |
| FT | 95.5 | 96.4 |
| RM@1 | 93.4 | 93.9 |
| RM@2 | 93.5 | 93.9 |
| RM@3 | 93.5 | 93.9 |

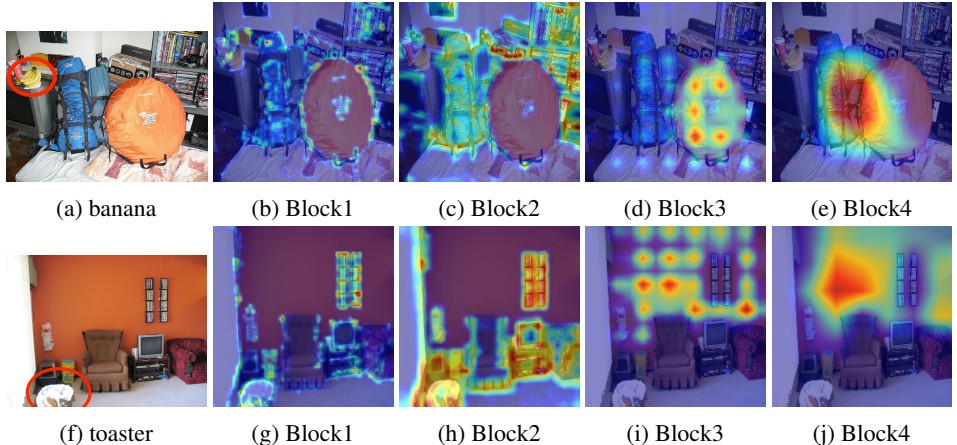

(a) banana    (b) Block1    (c) Block2    (d) Block3    (e) Block4

(f) toaster    (g) Block1    (h) Block2    (i) Block3    (j) Block4

Figure 2: Class activation maps with EigenCAM [25] at each block of ResNet101x3 trained with proposed method, which support class label (banana, suitcase, toaster).

activation maps support for are presented in leftmost column. As can be seen from original images, the corresponding objects are unclear and partially occluded. The activation maps show they are only captured by lower blocks.

Table 3: 8 Classes with largest mean scale values.

| Class | **banana** | toaster | **orange** | apple | **microwave** | oven | bird | **brocoli** |
|---|---|---|---|---|---|---|---|---|
| Scale value $\alpha$ | 9.09E-10 | 7.91E-10 | 7.47E-10 | 6.92E-10 | 6.70E-10 | 6.34E-10 | 6.12E-10 | 6.07E-10 |

**Mixup Scale Values**   Table 3 shows 10% largest class-wise mean scale values out of 80 classes, extracted from the inference results of the proposed method on MS-COCO validation set with ResNet101. In short, classes with large values were found to have shown poor performance in baseline (LP). They are marked in bold which belong in Figure 3(b). This implies that the greater the value, the more support can be provided to classes with low performance.

## 4 Conclusion and Future Work

This paper explores the effect of using intermediate representation, which can deal with hard samples in multi-label classification. Mixing up intermediate representation can maintain the efficiency which lies in linear probing and properly captures spatial information that could be lost when a feature is extracted in deep backbone networks. We provide methods to effectively refine the raw hidden representation into a form that could be interpolated with features extracted from a backbone network. The proposed method requires very few additional parameters from linear probing and achieves comparable performance to full fine-tuning. We confirm that intermediate representation can act as a hint for exploiting spatial information from any part of deep neural networks. We suggest future works could explore methods for instance-wise training using intermediate representation from different layers, which could pave the way for adaptive linear probing.

## Acknowledgments

This work was supported by Institute of Information & communications Technology Planning & Evaluation (IITP) grant funded by Korea government (MSIT) [No. 2021-0-00907, Development of Adaptive and Lightweight Edge-Collaborative Analysis Technology for Enabling Proactively Immediate Response and Rapid Learning, 90%] and [No. 2019-0-00075, Artificial Intelligence Graduate School Program (KAIST), 10%].

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

# A   Appendix

## A.1   Investigation

Here, the architectures are trained on ImageNet21k (or ImageNet22k), and the linear classifier is trained for 50 epochs with SGD optimizer of momentum 0.9. For learning rate, we use one cycle learning rate scheduler with maximum learning rate 0.1 following recent work on MLC in MS-COCO [29].

**Investigation.**   In Figure 3, we list the best and worst 10% performing classes for various backbone networks, which shows consistency throughout all networks. Then, for each class in poor performance, we arrange images that contain it as a true label according to model output scores. They often contain small objects that are hard to recognize and partially occluded. Figure 4 is one of the examples that shows the object labels take part in different number of areas. Further examples can be found in Appendix A.

**Hard Samples in MLC.**   As we visually explore samples throughout each class, typical hard samples have a small part of an object co-occurred with other larger-sized objects. Previous works [12] that analyze intermediate feature maps can resort to CNN stimulus in final output proved a high correlation between average precision and average part size of each class. For example, Figure 5 contains multiple labels of various sizes. With explainable visualization method [4], we look through how each layer reacts to the class label "banana" differently. As expected, the small-sized object reacts the most on intermediate layers ((d) and (h)), but this reaction disappears on the last layer.

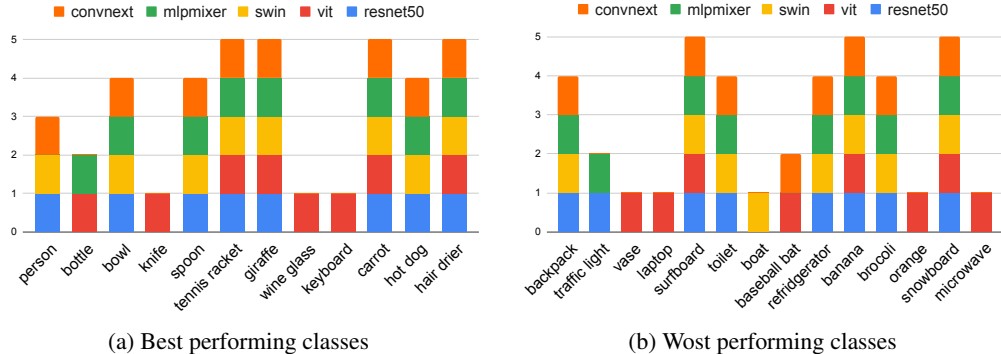

(a) Best performing classes

(b) Wost performing classes

Figure 3: Class-wise performance per backbone out of 80 classes in total. The backbone networks mostly share similar result. We note that these classes ranged widely in terms of class size.

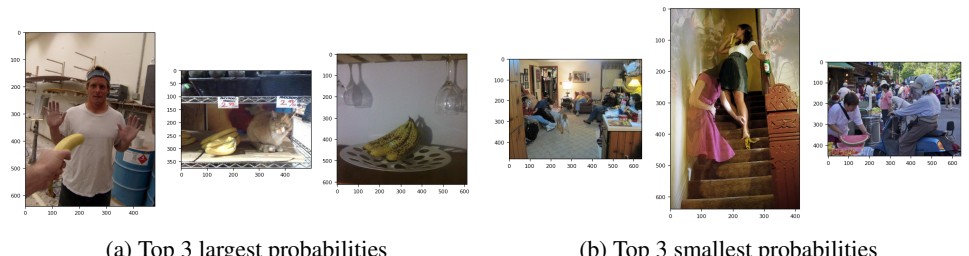

(a) Top 3 largest probabilities

(b) Top 3 smallest probabilities

Figure 4: Images with 'Banana' class whose predicted probability is either the largest(a) or smallest(b).

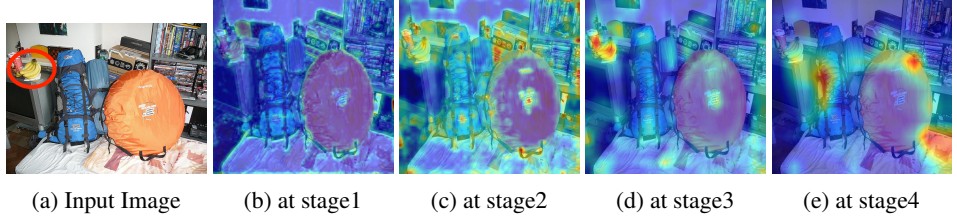

(a) Input Image     (b) at stage1     (c) at stage2     (d) at stage3     (e) at stage4

Figure 5: Original Image of class labels [backpack, banana, bed, tv, book, bottle, person] and GradCAM++ results [4].

Figure 5 shows that this complicated sample means many false negatives vanish as passing large and deep backbone networks. Therefore, the performance has room for improvement by capturing the hard samples, and intermediate representations can be the key to the solution.

## A.2 Experiment Setting

**Datasets.** We use backbone network pre-trained on ImageNet-21K or ImageNet-22k [28, 8] which contains upto 14.2 milllion images from 22K classes. For evaluation of proposed method, we use MS-COCO [22], Pascal-VOC [11] which contain 2.5 million annotated objects in 328k images and 24,640 annotated objects in 9,963 images, respectively.

**Metrics.** For multi-label classification task, the labels are predicted as positive if the confidences of them are greater than 0.5. We compute class-wise precision (CP), recall (CR), F1 (CF1) and overall precision (OP), recall (OR) and F1 (OF1). Those not reported in main section are reported in A. Recent works on MLC [6, 29, 21, 35, 5] report mean average precision (mAP) as main metric. Hence, we follow the convention.

Table 4: Classwise AP difference between LP and RM on MS-COCO dataset.

| class | snowboard | surfboard | car | toilet | refrigerator | broccoli | traffic light | truck | banana | cell phone |
|---|---|---|---|---|---|---|---|---|---|---|
| LP | 13.53 | 37.51 | 40.08 | 45.49 | 45.71 | 51.45 | 56.31 | 57.69 | 60.60 | 60.96 |
| RM | 16.37 | 47.08 | 40.44 | 46.82 | 45.21 | 54.29 | 61.39 | 60.60 | 62.95 | 62.43 |
| difference | 2.83 | 9.57 | 0.35 | 1.33 | -0.50 | 2.85 | 5.08 | 2.91 | 2.35 | 1.47 |

| class | pizza | sink | boat | train | baseball bat | toothbrush | backpack | bed | mouse | chair |
|---|---|---|---|---|---|---|---|---|---|---|
| LP | 61.50 | 62.37 | 63.19 | 64.57 | 64.95 | 65.01 | 65.64 | 66.36 | 66.92 | 67.00 |
| RM | 63.30 | 63.08 | 65.75 | 65.20 | 66.69 | 65.93 | 66.99 | 66.98 | 69.98 | 68.88 |
| difference | 1.80 | 0.71 | 1.56 | 0.63 | 1.74 | 0.92 | 1.35 | 0.62 | 3.06 | 1.88 |

| class | tv | toaster | microwave | orange | cat | parking meter | suitcase | sandwich | laptop | potted plant |
|---|---|---|---|---|---|---|---|---|---|---|
| LP | 67.25 | 68.26 | 68.99 | 69.29 | 69.67 | 70.34 | 70.96 | 71.75 | 73.69 | 74.04 |
| RM | 68.82 | 69.38 | 74.1 | 73.72 | 71.08 | 70.89 | 74.44 | 75.62 | 74.26 | 75.19 |
| difference | 1.57 | 1.12 | 5.11 | 4.43 | 1.41 | 0.55 | 3.48 | 3.87 | 0.57 | 1.15 |

| class | oven | donut | airplane | bench | fire hydrant | bear | bus | dog | fork | kite |
|---|---|---|---|---|---|---|---|---|---|---|
| LP | 74.47 | 74.69 | 75.47 | 75.9 | 76.11 | 76.38 | 76.67 | 77.02 | 77.28 | 78.97 |
| RM | 76.00 | 76.48 | 79.56 | 78.44 | 79.28 | 78.78 | 77.67 | 79.84 | 78.13 | 80.50 |
| difference | 1.53 | 1.80 | 4.10 | 2.54 | 3.17 | 2.40 | 1.00 | 2.81 | 0.85 | 1.54 |

| class | zebra | dining table | umbrella | horse | apple | scissors | remote | bird | cow | knife |
|---|---|---|---|---|---|---|---|---|---|---|
| LP | 79.47 | 80.07 | 80.09 | 80.16 | 80.65 | 80.93 | 81.19 | 81.95 | 82.67 | 82.81 |
| RM | 79.66 | 83.12 | 81.92 | 81.19 | 81.12 | 82.81 | 85.26 | 82.7 | 84.79 | 85.13 |
| difference | 0.18 | 3.05 | 1.84 | 1.03 | 0.46 | 1.88 | 4.07 | 0.75 | 2.12 | 2.32 |

| class | baseball glove | skateboard | clock | vase | frisbee | book | cake | handbag | motorcycle | couch |
|---|---|---|---|---|---|---|---|---|---|---|
| LP | 83.18 | 83.26 | 83.53 | 86.15 | 87.17 | 87.79 | 88.97 | 89.03 | 90.52 | 90.66 |
| RM | 83.83 | 85.43 | 85.47 | 88.5 | 88.44 | 88.44 | 90.03 | 90.24 | 93.35 | 91.71 |
| difference | 0.65 | 2.17 | 1.93 | 2.36 | 1.27 | 0.64 | 1.05 | 1.21 | 2.83 | 1.05 |

| class | tie | keyboard | wine glass | bicycle | sports ball | bottle | elephant | stop sign | cup | sheep |
|---|---|---|---|---|---|---|---|---|---|---|
| LP | 91.75 | 92.88 | 93.08 | 93.1 | 94.38 | 94.53 | 94.68 | 94.75 | 95.65 | 95.87 |
| RM | 93.80 | 93.88 | 94.45 | 94.59 | 95.45 | 95.21 | 95.71 | 95.79 | 96.03 | 97.04 |
| difference | 2.04 | 1.01 | 1.36 | 1.49 | 1.07 | 0.69 | 1.03 | 1.04 | 0.37 | 1.16 |

| class | hot dog | teddy bear | skis | person | spoon | bowl | hair drier | giraffe | carrot | tennis racket |
|---|---|---|---|---|---|---|---|---|---|---|
| LP | 95.94 | 96.03 | 96.11 | 96.73 | 97.54 | 97.64 | 98.57 | 98.58 | 99.04 | 99.16 |
| RM | 96.43 | 97.56 | 96.61 | 96.96 | 98.34 | 97.71 | 98.25 | 98.57 | 99.18 | 99.42 |
| difference | 0.49 | 1.54 | 0.50 | 0.23 | 0.80 | 0.07 | -0.32 | -0.01 | 0.14 | 0.26 |

**Implementation Details.** We use SGD optimizer with momentum 0.9, MultiLabelSoftMarginLoss and one-cycle learning rate scheduler. The chosen maximum learning rate is 0.1 by default, except for fine-tuning parameters which are set to 0.01. We checked for convergence in order to figure out appropriate length of training epochs and 30 was enough to show the gain. In addition, the aim of transfer learning is to achieve efficacy in training. Therefore, all the results reported are best results during training for 30 epochs. We tried using augmentation methods such as cutout [9] and RandAugment [7], but decided to omit them for the effect was minimal due to short training time.

## A.3 Classwise AP

Table 4 shows the classwise AP difference between LP and RM on MS-COCO dataset. We verify that the proposed RM method outperforms LP method in most classes. Especially, we can observe that certain minor classes are significantly improved.

## A.4 Related Work

To improve the transferability against a target domain, the methods for transfer learning have been evolving into two main directions: (1) *Architecture-based Transfer Learning*, which utilizes the weight parameters of pre-trained networks, and (2) *Feature-based Transfer Learning*, which designs new objective functions to transfer the output knowledge of pre-trained networks.

**Architecture-based Transfer Learning** Architecture-based transfer learning approaches focus on finding the optimal backbone model and its representation for a given target task. Alain and Bengio [1] use an auxiliary linear classifier to evaluate intermediate layers of black box deep neural networks. Kumar et al. [20] show accuracy trade-off between in-distribution and out-of-distribution (OOD) of target datasets with fine-tuning and suggest a fine-tuning strategy for OOD distribution with initialization of linear-probed head. Also, greedy selection of pre-trained models and averaging their weights can show similar results as the logit ensembling method concerning the flatness of loss and confidence (Wortsman et al. [34]). Kirichenko et al. [17] use linear probing linear classifier as a tool to separate pre-trained networks from spurious features and focus on core features.

**Feature-based Transfer Learning**   The indispensable key to the success of deep neural networks is huge parameters. However, the increase in performance cannot be achieved without increasing the training and testing time. Therefore, several methods have been developed to save on training costs with pre-trained networks. Knowledge transfer aims to make the network smaller while maintaining the performance of pre-trained large neural networks. For this purpose, the knowledge from the teacher network is distilled into a smaller student network. Romero et al. [30] use interlayer values used as clues to enable knowledge distillation for deeper and sparser student networks. Yim et al. [36] present a hint-based transfer learning technique in which the hint is generated from features of two intermediate layers of the teacher network. Evci et al. [10] propose a TL method that selects intermediate features of the backbone network to train the classifier head. Likewise, some recent work focuses on measuring values with pre-trained weights for matching source models and target data that do not require fine-tuning [16, 27, 26, 37].

**Multi-label Classification**   Many recent advanced deep learning architectures have been developed for MLC tasks [23]. Wu et al. [35] deal with tail class instances in long-tailed distribution and propose re-sampling and re-weight-based approach to improve training. However, most ML approaches utilize both image and label embedding to effectively learn semantic label dependency and relevance between them. Wang et al. [32] use RNN to learn high-order label dependencies in an image. Chen et al. [5] build a directed graph over object labels and employ GCN to model the label correlations, and learned label representation is mapped to the classifier. Wang et al. [33] add a lateral connection between GCN classifier and CNN backbone at intermediate layers so that label information can be injected into the backbone. Furthermore, more recent approaches use attention models such as GAT or Transformer, where attention is computed on labels and images. Cheng et al. [6], Yuan et al. [38] introduce Transformer models for MLC tasks in which self-attention is applied to image representation. Some researchers [40, 29, 21] use learnable label embedding initialized by the pre-trained model to compute attention with the input image.

