# OpenReview forum: "Layover Intermediate Layer for Multi-Label Classification in Efficient Transfer Learning"
_NeurIPS.cc/2022/Workshop/HITY — HITY Workshop NeurIPS 2022_

### Official Review · Reviewer_DELq · 2022-10-17

**Rating:** 1
**Confidence:** 4

**Review:**

This manuscript introduces a new approach for transfer learning of multi-label classification models. The paper is clearly written and presents compelling results on the efficacy of the method, especially on hard instances where images contain multiple objects with significant size differences.

---

### Official Review · Reviewer_YHje · 2022-10-19

**Rating:** 1
**Confidence:** 3

**Review:**

Introduces a new multilabel classification technique for hard instances. Does some preliminary tests against baselines, and results seem promising.

---

### Decision · Program_Chairs · 2022-10-20

Accept